# IgA Vasculitis: Influence of *CD40, BLK* and *BANK1* Gene Polymorphisms

**DOI:** 10.3390/jcm11195577

**Published:** 2022-09-22

**Authors:** Joao Carlos Batista Liz, Fernanda Genre, Verónica Pulito-Cueto, Sara Remuzgo-Martínez, Diana Prieto-Peña, Ana Márquez, Norberto Ortego-Centeno, María Teresa Leonardo, Ana Peñalba, Javier Narváez, Luis Martín-Penagos, Lara Belmar-Vega, Cristina Gómez-Fernández, José A. Miranda-Filloy, Luis Caminal-Montero, Paz Collado, Diego De Árgila, Patricia Quiroga-Colina, Esther F. Vicente-Rabaneda, Ana Triguero-Martínez, Esteban Rubio, Manuel León Luque, Juan María Blanco-Madrigal, Eva Galíndez-Agirregoikoa, Javier Martín, Oreste Gualillo, Ricardo Blanco, Santos Castañeda, Miguel A. González-Gay, Raquel López-Mejías

**Affiliations:** 1Research Group on Genetic Epidemiology and Atherosclerosis in Systemic Diseases and in Metabolic Bone Diseases of the Musculoskeletal System, IDIVAL, 39011 Santander, Spain; 2Division of Rheumatology, Hospital Universitario Marqués de Valdecilla, 39008 Santander, Spain; 3Instituto de Parasitología y Biomedicina ‘López-Neyra’, CSIC, PTS Granada, 18016 Granada, Spain; 4Department of Medicine, Universidad de Granada, 18071 Granada, Spain; 5Division of Paediatrics, Hospital Universitario Marqués de Valdecilla, 39008 Santander, Spain; 6Division of Rheumatology, Hospital Universitario de Bellvitge, 08907 Barcelona, Spain; 7Division of Nephrology, Hospital Universitario Marqués de Valdecilla, IDIVAL-REDINREN, 39008 Santander, Spain; 8Division of Dermatology, Hospital Universitario Marqués de Valdecilla, 39008 Santander, Spain; 9Division of Rheumatology, Hospital Universitario Lucus Augusti, 27003 Lugo, Spain; 10Internal Medicine Department, Hospital Universitario Central de Asturias, Instituto de Investigación Sanitaria del Principado de Asturias (ISPA), 33011 Oviedo, Spain; 11Division of Rheumatology, Hospital Universitario Severo Ochoa, 28911 Madrid, Spain; 12Division of Dermatology, Hospital Universitario de La Princesa, 28006 Madrid, Spain; 13Division of Rheumatology, Hospital Universitario de La Princesa, IIS-Princesa, 28006 Madrid, Spain; 14Department of Rheumatology, Hospital Universitario Virgen del Rocío, 41013 Sevilla, Spain; 15Division of Rheumatology, Hospital Universitario de Basurto, 48013 Bilbao, Spain; 16SERGAS (Servizo Galego de Saude) and IDIS (Instituto de Investigación Sanitaria de Santiago), NEIRID Lab (Neuroendocrine Interactions in Rheumatology and Inflammatory Diseases), Research Laboratory 9, Hospital Clínico Universitario de Santiago, 15706 Santiago de Compostela, Spain; 17Department of Medicine and Psychiatry, Universidad de Cantabria; 39005, Santander, Spain; 18Cardiovascular Pathophysiology and Genomics Research Unit, Faculty of Health Sciences, School of Physiology, University of the Witwatersrand, Johannesburg 2050, South Africa

**Keywords:** IgA vasculitis, Henoch–Schönlein purpura, *CD40*, *BLK*, *BANK1*, polymorphisms

## Abstract

*CD40, BLK* and *BANK1* genes involved in the development and signaling of B-cells are identified as susceptibility loci for numerous inflammatory diseases. Accordingly, we assessed the potential influence of *CD40, BLK* and *BANK1* on the pathogenesis of immunoglobulin-A vasculitis (IgAV), predominantly a B-lymphocyte inflammatory condition. Three genetic variants within *CD40* (rs1883832, rs1535045, rs4813003) and *BLK* (rs2254546, rs2736340, rs2618476) as well as two *BANK1* polymorphisms (rs10516487, rs3733197), previously associated with inflammatory diseases, were genotyped in 382 Caucasian patients with IgAV and 955 sex- and ethnically matched healthy controls. No statistically significant differences were observed in the genotype and allele frequencies of *CD40, BLK* and *BANK1* when IgAV patients and healthy controls were compared. Similar results were found when *CD40, BLK* and *BANK1* genotypes or alleles frequencies were compared between patients with IgAV stratified according to the age at disease onset or to the presence/absence of gastrointestinal or renal manifestations. Moreover, no *CD40, BLK* and *BANK1* haplotype differences were disclosed between patients with IgAV and healthy controls and between patients with IgAV stratified according to the clinical characteristics mentioned above. Our findings indicate that *CD40, BLK* and *BANK1* do not contribute to the genetic background of IgAV.

## 1. Introduction

B-lymphocytes are key cells for an effective immune response, mainly because they produce immunoglobulins (Igs) [1]. Cluster of differentiation 40 (CD40), a glycoprotein expressed on the surface of B-cells, participates in the activation [2], survival, proliferation and differentiation of these lymphocytes and in the isotype switching of Igs [3]. B-lymphoid kinase (BLK) and B-cell scaffold protein with ankyrin repeats 1 (BANK1) are components of the B-cells’ signalosome [4]. In this regard, BLK is a src family nonreceptor tyrosine kinase [5] with a relevant role in the development and receptor signaling of B-cells [6], whereas BANK1 is an adaptor/scaffold that participates in B-cell activation and signalization [7,8]. Interestingly, *CD40* [9,10,11,12,13], *BLK* [7,14,15,16,17,18] and *BANK1* [7,8,18,19] genes are identified as susceptibility *loci* for several inflammatory diseases. Likewise, *CD40* and *BLK* variants are known as susceptibility factors for different forms of vasculitis, specifically for the development of ischemic manifestations in patients with giant cell arteritis [20,21] and for Kawasaki disease [13,22], supporting the relevance of B-cell activation in the pathophysiology of both vasculitides.

Immunoglobulin-A vasculitis (IgAV), or Henoch–Schönlein purpura (HSP), is a small-sized blood vasculitis, more common in children and rarer but more severe in adults [23,24,25]. Besides the classic clinical triad of palpable purpura, arthralgias/arthritis and gastrointestinal (GI) tract involvement [26,27], renal damage is also presented in patients with IgAV, constituting the most serious complication of this vasculitis [28,29]. Abnormal IgA deposits in the vessel walls are the characteristic pathophysiologic feature of IgAV [23], supporting the theory that this vasculitis is predominantly a B-cell disease. The etiology of IgAV has not been completely elucidated. Nevertheless, numerous pieces of evidence support the claim that genetics is crucial in the pathogenesis of this condition [30,31,32].

Accordingly, we aimed to determine, for the first time, the potential influence of *CD40, BLK* and *BANK1* on the pathogenesis of IgAV. For this purpose, eight polymorphisms (three within *CD40*, three within *BLK* and two within *BANK1*) were genotyped in the largest series of Caucasian IgAV patients ever assessed for genetic studies.

## 2. Materials and Methodology

### 2.1. Study Population

The study group of the present work encompassed a total of 382 unrelated patients who fulfilled Michel et al.’s criteria [33] and/or the American College of Rheumatology’s classification criteria [34] for IgAV-HSP and/or the 2012 revised International Chapel Hill Consensus Conference Nomenclature [35] definition for IgAV. All these patients were Spaniards of European ancestry and were recruited in the following healthcare centers: Hospital Universitario Marqués de Valdecilla (Santander), Hospital Universitario Clínico San Cecilio (Granada), Hospital Universitario de Bellvitge (Barcelona), Hospital Universitario Lucus Augusti (Lugo), Hospital Universitario Central de Asturias (Oviedo), Hospital Universitario Severo Ochoa and Hospital Universitario de La Princesa (Madrid), Hospital Universitario Virgen del Rocío (Sevilla) and Hospital Universitario de Basurto (Bilbao). A description of the main clinical characteristics of the patients with IgAV included in this study is presented in Table 1. GI manifestation was considered present if bowel angina and GI bleeding were observed as previously described [31]. Renal manifestations were defined to be present if hematuria, proteinuria or nephrotic syndrome at any time over the clinical course of the disease and/or renal sequelae (persistent renal involvement) at last follow up was disclosed [31].

In addition, 955 sex- and ethnically matched, unrelated individuals without a history of cutaneous vasculitis or any other autoimmune disease from Hospital Universitario Marqués de Valdecilla (Santander) and National DNA Bank Repository (Salamanca) were also included in our work as healthy controls.

All subjects gave their informed consent to be included in the study. The procedures followed were in accordance with the ethical standards of the approved guidelines and regulations, in accordance with the Declaration of Helsinki. All experimental protocols were approved by the local ethics committees of each participant hospital (approval code 15/2012 and date of approval 11 May 2012).

### 2.2. Selection of Single-Nucleotide Polymorphisms and Genotyping Method

Three polymorphisms within *CD40* (rs1883832, rs1535045 and rs4813003) and *BLK* (rs2254546, rs2736340, rs2618476) genes as well as two genetic variants within *BANK1* (rs10516487 and rs3733197) were selected in this study. These eight specific variants were selected considering that they were related to numerous inflammatory disorders [7,8,9,10,11,12,13,14,15,16,17,18,19]. In addition, potential functional consequences were previously proposed for some of these polymorphisms [8,18,19,36].

DNA from all the IgAV patients and healthy controls included in the study was extracted from peripheral blood samples using the REALPURE “SSS” kit (RBME04, REAL, Durviz S.L., Spain). All individuals were genotyped for the eight genetic variants mentioned above using predesigned TaqMan genotyping probes (C__11655919_20 for rs1883832, C___1260189_10 for rs1535045, C___1260313_20 for rs4813003, C__16036468_10 for rs2254546, C___1886931_30 for rs2736340, C__16036467_10 for rs2618476, C____313748_30 for rs10516487 and C___1793403_1_ for rs3733197) in a QuantStudio^TM^ 7 Flex real-time polymerase chain reaction system, according to the conditions recommended by the manufacturer (Applied Biosystems, Foster City, CA, USA).

Negative controls and duplicate samples were incorporated in our study to check the genotyping accuracy.

### 2.3. Statistical Analysis 

*CD40* rs1883832, *CD40* rs1535045, *CD40* rs4813003, *BLK* rs2254546, *BLK* rs2736340, *BLK* rs2618476, *BANK1* rs10516487 and *BANK1* rs3733197 genotypes were examined for deviation from the Hardy–Weinberg equilibrium (HWE).

To test for association, we compared *CD40, BLK* and *BANK1* frequencies between patients with IgAV and healthy controls as well as between patients with IgAV stratified according to specific clinical characteristics of the disease (age at the disease onset or presence/absence of GI or renal manifestations).

*CD40, BLK* and *BANK1* variants were evaluated independently. Genotype and allele frequencies were calculated and compared between the groups mentioned above using chi-square test or Fisher test. Strength of association was estimated using odds ratio (OR) and 95% confidence intervals (CIs).

Then, we carried out the allelic combination (haplotype) analysis for the three *CD40* genetic variants studied as well as for the three *BLK* polymorphisms assessed and the two *BANK1* variants evaluated. Haplotype frequencies were calculated by the Haploview v4.2 software (http://broad.mit.edu/mpg/haploview) (accessed on 20 September 2022) and compared between the groups mentioned above using chi-square test. The strength of association was estimated by OR and 95% CI.

We considered results with *p*-values <0.05 as statistically significant.

All statistical analyses were conducted with the STATA statistical software 12/SE (Stata Corp., College Station, TX, USA).

## 3. Results

No deviation from HWE was detected for *CD40* rs1883832, *CD40* rs1535045, *CD40* rs4813003, *BLK* rs2254546, *BLK* rs2736340, *BLK* rs2618476, *BANK1* rs10516487 and *BANK1* rs3733197 in healthy controls.

The genotyping success rate was greater than 98% for the eight polymorphisms evaluated in this study.

Both genotype and allele frequencies of *CD40, BLK* and *BANK1* variants assessed were in accordance with those reported in the 1000 Genomes Project (http://www.internationalgenome.org/) (accessed on 20 September 2022) for European populations.

### 3.1. CD40, BLK and BANK1 Genotype and Allele Frequencies in Patients with IgAV and Healthy Controls

Genotype and allele frequencies of *CD40, BLK* and *BANK1* polymorphisms assessed independently were compared between patients with IgAV and healthy controls.

In this respect, similar genotype and allele *CD40, BLK* and *BANK1* frequencies were observed in patients with IgAV when compared to healthy controls (Table 2).

### 3.2. CD40, BLK and BANK1 Genotype and Allele Frequencies in Patients with IgAV Stratified according to Specific Clinical Characteristics of the Disease

Subsequently, we evaluated whether differences in the genotype and allele frequencies of each *CD40, BLK* and *BANK1* polymorphism could exist between IgAV patients stratified according to specific clinical characteristics of the disease.

Since IgAV is often a benign and self-limited pathology in children and a more severe condition in adults, we analyzed potential differences in *CD40, BLK* and *BANK1* genotype and allele frequencies between patients with IgAV stratified according to the age at disease onset. As shown in Table 3, no statistically significant *CD40, BLK* and *BANK1* differences were found in children when compared to adults.

Moreover, we analyzed *CD40, BLK* and *BANK1* genotype and allele frequencies between patients with IgAV stratified according to the presence/absence of GI or renal manifestations. In this regard, similar *CD40, BLK* and *BANK1* frequencies were observed when IgAV patients were stratified according to the presence/absence of GI manifestations (Table 3). This was also the case when patients with IgAV who developed renal manifestations were compared to those without these complications (Table 3).

### 3.3. CD40, BLK and BANK1 Haplotype Analyses

Moreover, we investigated whether *CD40, BLK* and *BANK1* haplotype frequencies differed between IgAV patients and controls as well as between IgAV patients stratified according to the specific clinical characteristics of the disease above mentioned.

In this regard, no statistically significant *CD40, BLK* and *BANK1* haplotypes differences were disclosed in patients with IgAV when compared to healthy controls (Table 4).

Likewise, CD40, BLK and BANK1 haplotype frequencies were similar between IgAV patients stratified according to the age at disease onset or to the presence/absence of GI or renal manifestations (Table 5).

## 4. Discussion

B-lymphocytes play essential functions in regulating immune responses [37], being rigorously regulated with respect to both development and activation [1]. Abnormalities in these processes contribute to the pathogenesis of autoimmune diseases [38]. Cumulative knowledge clearly demonstrated that CD40, BLK and BANK1 are key proteins involved in the development and signaling of B-cells [2,3,4,5,6,7,8]. Additionally, *CD40, BLK* and *BANK1* genes are identified as shared susceptibility *loci* for several inflammatory diseases [7,8,9,10,11,12,13,14,15,16,17,18,19].

Taking these considerations into account, we evaluated whether *CD40, BLK* and *BANK1* are also implicated in the pathogenesis of IgAV, predominantly a B-cell inflammatory condition, involving small blood vessels. For that purpose, three polymorphisms within *CD40* and *BLK* as well as two genetic *BANK1* variants, previously associated with several inflammatory diseases [7,8,9,10,11,12,13,14,15,16,17,18,19], were evaluated in the largest series of Caucasian patients with IgAV ever assessed for genetic studies. Some of these variants also exhibit different functional consequences [8,18,19,36]. In particular, the *CD40* rs1883832C allele influences the translational efficiency of nascent CD40 mRNA transcripts, resulting in elevated CD40 levels [8]. *BLK* rs2736340 is in tight linkage disequilibrium with rs13277113 (D’ = 1, r^2^ = 0.99 in Europeans), wherein the A allele is associated with lower levels of mRNA *BLK* [36]. Finally, *BANK1* rs10516487 leads to a substitution of Arg to His at amino-acid position 61 of BANK1 protein, whereas *BANK1* rs3733197 causes an Ala to Thr substitution at amino-acid position 383 (creating a site for threonine kinases that affects the B-cell signaling) [18,19]. Interestingly, our findings revealed no influence of *CD40, BLK* and *BANK1* on IgAV susceptibility when we studied each of the polymorphisms separately or together, conforming haplotypes. Several studies described the influence of different polymorphisms on the increased risk of nephritis or GI disease in IgAV [39,40,41,42]. Accordingly, we also evaluated the potential association of *CD40, BLK* and *BANK1* with the increased risk of nephritis or GI complications in our study. Nevertheless, our results do not support a role of *CD40, BLK* and *BANK1* variants with clinical features of IgAV, suggesting that these genes do not contribute to IgAV severity. Notwithstanding, our results do not exclude the potential implication of other polymorphisms related to B-cells in the pathogenesis of IgAV. Consequently, further studies are needed to clarify this issue.

Shared molecular mechanisms among different vasculitides have been described [43,44]. However, as observed in our series of IgAV, no association of *BANK1* rs10516487 and *BANK1* rs3733197 variants with the susceptibility and clinical expression of patients with giant cell arteritis [45], a primary systemic vasculitis that involves large- and middle-sized blood vessels in people older than 50 years, was disclosed. It was also the case for the potential implication of *BLK* rs2736340 and *BANK1* rs10516487 in Takayasu arteritis [46], another primary large-vessel vasculitis that involves mainly young individuals. In contrast, a potential influence of *CD40* rs1883832 [20] and *BLK* rs2736340 [21] polymorphisms was previously reported on the development of ischemic manifestations in patients with giant cell arteritis. Similarly, *CD40* rs4813003 [13], *BLK* rs2254546 [13], *BLK* rs2736340 [22] and *BLK* rs2618476 [22] were identified as susceptibility markers for Kawasaki disease, a vasculitis affecting small- and medium-sized arteries.

Our findings suggest that IgAV may not be a state of increased B-cell activation, pointing to IgAV as a different entity from other types of vasculitis. In keeping with our results, genome-wide association studies in IgA nephritis, which is pathophysiologically similar to IgAV [47,48], have not identified *CD40, BLK* and *BANK1* genes as significant players in the pathogenesis of the disease [49,50,51]. With respect to this, genes affecting the mucosal immune defense, having an impact on IgA production by plasma cells in mucosa and previously reported as susceptibility *loci* in IgA nephropathy [51], may also be implicated in the pathogenesis of IgAV, and further studies assessing this issue would be of potential interest.

## 5. Conclusions

In summary, although complex genetic interactions appear to be involved in the pathogenesis of IgAV, based on a large series of patients, we could not observe a contribution of the *CD40, BLK* and *BANK1* genes to the genetic network underlying this small-vessel vasculitis. Further studies should be performed to fully explore the role of B-cells in the pathogenesis of IgAV.

## Figures and Tables

**Table 1 jcm-11-05577-t001:** Main clinical characteristic of the 382 IgAV patients recruited in our study.

	% (n)
Children (age ≤20 years)/adults (age >20 years) (n)	296/86
Percentage of females	47.4
Age at disease onset (years, median [IQR])	7 [5,6,7,8,9,10,11,12,13,14,15,16,17,18,19]
Duration of follow up (years, median [IQR])	1 [1,2,3]
Palpable purpura and/or maculopapular rash	100 (382)
Arthralgia and/or arthritis	55.5 (212)
GI manifestations (if “a” and/or “b”)	53.1 (203)
(a) Bowel angina	50.3 (192)
(b) GI bleeding	16.8 (64)
Renal manifestations (if any of the following characteristics)	36.1 (138)
(a) Hematuria ^1^	34.8 (133)
(b) Proteinuria ^1^	33.5 (128)
(c) Nephrotic syndrome ^1^	5.8 (22)
(d) Renal sequelae (persistent renal involvement) ^2^	6.8 (26)

IgAV: IgA vasculitis; IQR: interquartile range; GI: gastrointestinal. ^1^ At any time over the clinical course of the disease; ^2^ at last follow up.

**Table 2 jcm-11-05577-t002:** *CD40, BLK* and *BANK1* genotype and allele frequencies in patients with IgAV and healthy controls.

		Change		Genotypes, % (n)			Allele Test^1^	
*Locus*	SNP	1/2	Sample Set	1/1	1/2	2/2	p	OR [95% CI]
*CD40*	rs1883832	C/T	IgAV patients	54.0 (205)	36.8 (140)	9.2 (35)	0.69	1.04 [0.86–1.26]
			Healthy controls	53.1 (507)	40.1 (383)	6.8 (65)		
	rs1535045	C/T	IgAV patients	53.6 (201)	39.5 (148)	6.9 (26)	0.45	1.08 [0.88–1.31]
			Healthy controls	56.8 (542)	36.0 (344)	7.2 (69)		
	rs4813003	C/T	IgAV patients	77.9 (292)	20.0 (75)	2.1 (8)	0.35	0.88 [0.68–1.15]
			Healthy controls	75.5 (721)	22.0 (210)	2.5 (24)		
*BLK*	rs2254546	G/A	IgAV patients	74.1 (278)	22.4 (84)	3.5 (13)	0.74	0.96 [0.75–1.22]
			Healthy controls	71.5 (683)	26.6 (254)	1.9 (18)		
	rs2736340	C/T	IgAV patients	63.0 (238)	31.7 (120)	5.3 (20)	0.60	0.95 [0.77–1.17]
			Healthy controls	60.0 (573)	35.8 (342)	4.2 (40)		
	rs2618476	T/C	IgAV patients	60.2 (228)	34.3 (130)	5.5 (21)	0.69	0.96 [0.78–1.18]
			Healthy controls	57.9 (553)	37.4 (357)	4.7 (45)		
*BANK1*	rs10516487	G/A	IgAV patients	52.5 (200)	40.2 (153)	7.3 (28)	0.80	0.98 [0.81–1.18]
			Healthy controls	51.2 (489)	41.8 (399)	7.0 (67)		
	rs3733197	G/A	IgAV patients	50.0 (188)	39.1 (147)	10.9 (41)	0.53	1.06 [0.88–1.28]
			Healthy controls	50.0 (478)	41.5 (396)	8.5 (81)		

IgAV: IgA vasculitis; SNP: single-nucleotide polymorphism; OR: odds ratio; CI: confidence interval.^1^ For the minor allele.

**Table 3 jcm-11-05577-t003:** *CD40, BLK* and *BANK1* genotype and allele frequencies in IgAV patients stratified according to clinical characteristics.

Locus	SNP	Children (Age ≤20 Years)		GI Manifestations ^1^		Renal Manifestations ^2^	
		Yes (n = 296)	No (n = 86)	Yes (n = 203)	No (n = 179)	Yes (n = 138)	No (n = 244)
*CD40*	rs1883832						
	CC	53.7 (159)	54.8 (46)	52.2 (105)	55.9 (100)	54.4 (74)	53.7 (131)
	CT	38.2 (113)	32.1 (27)	39.3 (79)	34.1 (61)	36.8 (50)	36.9 (90)
	TT	8.1 (24)	13.1 (11)	8.5 (17)	10.0 (18)	8.8 (12)	9.4 (23)
	C	72.8 (431)	70.8 (119)	71.9 (289)	72.9 (261)	72.8 (198)	72.1 (352)
	T	27.2 (161)	29.2 (49)	28.1 (113)	27.1 (97)	27.2 (74)	27.9 (136)
	rs1535045						
	CC	54.3 (158)	51.2 (43)	50.3 (100)	57.4 (101)	50.4 (68)	55.4 (133)
	CT	38.1 (111)	44.0 (37)	42.2 (84)	36.4 (64)	41.5 (56)	38.3 (92)
	TT	7.6 (22)	4.8 (4)	7.5 (15)	6.2 (11)	8.1 (11)	6.3 (15)
	C	73.4 (427)	73.2 (123)	71.4 (284)	75.6 (266)	71.1 (192)	74.6 (358)
	T	26.6 (155)	26.8 (45)	28.6 (114)	24.4 (86)	28.9 (78)	25.4 (122)
	rs4813003						
	CC	77.0 (224)	80.9 (68)	75.2 (152)	80.9 (140)	81.7 (112)	75.6 (180)
	CT	21.3 (62)	15.5 (13)	21.8 (44)	17.9 (31)	16.1 (22)	22.3 (53)
	TT	1.7 (5)	3.6 (3)	3.0 (6)	1.2 (2)	2.2 (3)	2.1 (5)
	C	87.6 (510)	88.7 (149)	86.1 (348)	89.9 (311)	89.8 (246)	86.8 (413)
	T	12.4 (72)	11.3 (19)	13.9 (56)	10.1 (35)	10.2 (28)	13.2 (63)
	rs2254546						
*BLK*	GG	74.5 (216)	73.0 (62)	77.9 (155)	69.9 (123)	73.3 (99)	74.6 (179)
	GA	22.1 (64)	23.5 (20)	18.6 (37)	26.7 (47)	23.7 (32)	21.7 (52)
	AA	3.4 (10)	3.5 (3)	3.5 (7)	3.4 (6)	3.0 (4)	3.7 (9)
	G	85.5 (496)	84.7 (144)	87.2 (347)	83.2 (293)	85.2 (230)	85.4 (410)
	A	14.5 (84)	15.3 (26)	12.8 (51)	16.8 (59)	14.8 (40)	14.6 (70)
	rs2736340						
	CC	63.1 (185)	62.4 (53)	64.8 (129)	60.9 (109)	69.1 (94)	59.5 (144)
	CT	31.4 (92)	32.9 (28)	30.2 (60)	33.5 (60)	27.2 (37)	34.3 (83)
	TT	5.5 (16)	4.7 (4)	5.0 (10)	5.6 (10)	3.7 (5)	6.2 (15)
	C	78.8 (462)	78.8 (134)	79.9 (318)	77.7 (278)	82.7 (225)	76.7 (371)
	T	21.2 (124)	21.2 (36)	20.1 (80)	22.3 (80)	17.3 (47)	23.3 (113)
	rs2618476						
	TT	60.2 (177)	60.0 (51)	61.9 (125)	58.2 (103)	65.0 (89)	57.4 (139)
	TC	33.3 (98)	37.6 (32)	34.1 (69)	34.5 (61)	32.1 (44)	35.6 (86)
	CC	6.5 (19)	2.4 (2)	4.0 (8)	7.3 (13)	2.9 (4)	7.0 (17)
	T	76.9 (452)	78.8 (134)	79.0 (319)	75.4 (267)	81.0 (222)	75.2 (364)
	C	23.1 (136)	21.2 (36)	21.0 (85)	24.6 (87)	19.0 (52)	24.8 (120)
*BANK1*	rs10516487						
	GG	52.5 (155)	52.3 (45)	51.0 (103)	54.2 (97)	50.0 (69)	53.9 (131)
	GA	40.0 (118)	40.7 (35)	41.1 (83)	39.1 (70)	40.6 (56)	39.9 (97)
	AA	7.5 (22)	7.0 (6)	7.9 (16)	6.7 (12)	9.4 (13)	6.2 (15)
	G	72.5 (428)	72.7 (125)	71.5 (289)	73.7 (264)	70.3 (194)	73.9 (359)
	A	27.5 (162)	27.3 (47)	28.5 (115)	26.3 (94)	29.7 (82)	26.1 (127)
	rs3733197						
	GG	48.6 (141)	54.6 (47)	47.5 (95)	52.8 (93)	46.0 (63)	52.3 (125)
	GA	41.4 (120)	31.4 (27)	43.5 (87)	34.1 (60)	41.6 (57)	37.7 (90)
	AA	10.0 (29)	14.0 (12)	9.0 (18)	13.1 (23)	12.4 (17)	10.0 (24)
	G	69.3 (402)	70.3 (121)	69.2 (277)	69.9 (246)	66.8 (183)	71.1 (340)
	A	30.7 (178)	29.7 (51)	30.8 (123)	30.1 (106)	33.2 (91)	28.9 (138)

IgAV: IgA vasculitis; SNP: single-nucleotide polymorphism; GI: gastrointestinal; ^1^ Bowel angina and/or gastrointestinal bleeding; ^2^ hematuria, proteinuria or nephrotic syndrome at any time over the clinical course of the disease and/or renal sequelae (persistent renal involvement) at last follow up.

**Table 4 jcm-11-05577-t004:** *CD40, BLK* and *BANK1* haplotype analysis in patients with IgAV and healthy controls.

CD40			p	OR [95% CI]
rs1883832	rs1535045	rs4813003		
C	C	C	-	Ref.
C	T	C	0.78	0.97 [0.76–1.23]
T	C	C	0.22	0.85 [0.65–1.11]
T	C	T	0.22	1.24 [0.87–1.81]
** *BLK* **			**p**	**OR [95% CI]**
**rs2254546**	**rs2736340**	**rs2618476**		
G	C	T	-	Ref.
G	T	C	0.10	1.22 [0.96–1.56]
A	C	T	0.05	1.35 [0.99–1.86]
A	T	C	0.44	0.85 [0.55–1.33]
** *BANK1* **			**p**	**OR [95% CI]**
**rs10516487**	**rs3733197**			
G	G		-	Ref.
A	A		0.90	1.01 [0.82–1.26]
G	A		0.06	0.74 [0.53–1.03]
A	G		0.66	0.92 [0.62–1.38]

IgAV: IgA vasculitis; OR: odds ratio; CI: confidence interval. Haplotypes of *CD40, BLK* and *BANK1* with a frequency higher than 5% are displayed in the table.

**Table 5 jcm-11-05577-t005:** *CD40, BLK* and *BANK1* haplotype analysis in IgAV patients stratified according to clinical characteristics.

			Age of Onset		Presence/Absence of GI Manifestations ^1^		Presence/Absence of Renal Manifestations ^2^	
*CD40*			p	OR [95% CI]	p	OR [95% CI]	p	OR [95% CI]
rs1883832	rs1535045	rs4813003						
C	C	C	-	Ref.	-	Ref.	-	Ref.
C	T	C	0.64	0.89 [0.55–1.48]	0.14	1.34 [0.89–2.04]	0.12	1.37 [0.90–2.09]
T	C	C	0.26	1.34 [0.78–2.35]	0.84	1.04 [0.66–1.65]	0.25	1.30 [0.81–2.06]
T	C	T	0.17	0.63 [0.31–1.34]	0.46	1.26 [0.65–2.47]	0.15	0.60 [0.26–1.25]
** *BLK* **			**p**	**OR [95% CI]**	**p**	**OR [95% CI]**	**p**	**OR [95% CI]**
rs2254546	rs2736340	rs2618476						
G	C	T	-	Ref.	-	Ref.	-	Ref.
G	T	C	0.31	1.30 [0.77–2.27]	0.73	0.93 [0.61–1.43]	0.07	0.66 [0.41–1.05]
A	C	T	0.41	1.32 [0.67–2.81]	0.44	0.81 [0.46–1.42]	0.80	1.07 [0.60–1.89]
A	T	C	0.51	0.78 [0.35–1.87]	0.05	0.50 [0.22–1.05]	0.66	0.85 [0.38–1.83]
** *BANK1* **			**p**	**OR [95% CI]**	**p**	**OR [95% CI]**	**p**	**OR [95% CI]**
rs10516487	rs3733197							
G	G		-	Ref.	-	Ref.	-	Ref.
A	A		0.96	0.99 [0.64–1.54]	0.82	1.04 [0.72–1.51]	0.24	1.25 [0.85–1.82]
G	A		0.40	1.33 [0.67–2.83]	0.72	1.01 [0.63–1.93]	0.51	1.20 [0.67–2.11]
A	G		0.57	1.26 [0.55–3.26]	0.28	1.43 [0.71–2.93]	0.82	1.08 [0.52–2.18]

IgAV: IgA vasculitis; GI: gastrointestinal; OR: odds ratio; CI: confidence interval. Haplotypes of *CD40, BLK* and *BANK1* with a frequency higher than 5% are displayed in the table; ^1^ bowel angina and/or gastrointestinal bleeding; ^2^ hematuria, proteinuria, or nephrotic syndrome at any time over the clinical course of the disease and/or renal sequelae (persistent renal involvement) at last follow up.

## Data Availability

All data generated or analyzed during this study are included in this published article.

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
