# Peer review of "IgA Vasculitis: Influence of CD40, BLK and BANK1 Gene Polymorphisms"

_jcm, 2022, doi:10.3390/jcm11195577_

Round 1
Reviewer 1 Report
In this study, the authors assessed the polymorphism of three genes involved in B cell development and activation on a population of patients suffering with IgA vasculitis. They found no difference for the variant frequencies in IgA vascultis patients compared to control group. This study is well designed and clear written. These patients are rare and the patient cohort is important.
The lack of positive results could be due to the choice of genes involved in B cell biology. Kyriluk et al in 2014 (Nature Genetics) showed in GWAS analysis on IgA nephropathy patients that most of genes were genes involved in mucosal immune defense (like ITGAX-ITGAM, VAV3, CARD9, TNFSF13), having an impact on IgA production by plasma cells in mucosa. Why did the authors not focus on genes directly impacting IgA+ plasmablasts/plasma cells as IgA production is linked to mucosa immunology? TNFSF13 (TACI) or BAFF genes, involved in later states of B cell differentiation would be more informative.
Author Response
Dear Reviewer 1,
Thank you very much for your feedback.
We have carefully read your interesting comments and modified our manuscript accordingly (please, see below).
In the following point-by-point reply, we have detailed the changes performed, which are shown in red in the revised version of our manuscript.
Point-by-point reply
In this study, the authors assessed the polymorphism of three genes involved in B cell development and activation on a population of patients suffering with IgA vasculitis. They found no difference for the variant frequencies in IgA vasculitis patients compared to control group. This study is well designed and clear written. These patients are rare and the patient cohort is important.
The lack of positive results could be due to the choice of genes involved in B cell biology. Kyriluk et al in 2014 (Nature Genetics) showed in GWAS analysis on IgA nephropathy patients that most of genes were genes involved in mucosal immune defense (like ITGAX-ITGAM, VAV3, CARD9, TNFSF13), having an impact on IgA production by plasma cells in mucosa. Why did the authors not focus on genes directly impacting IgA+plasmablasts/plasma cells as IgA production is linked to mucosa immunology? TNFSF13 (TACI) or BAFF genes, involved in later states of B cell differentiation would be more informative.
Response: We greatly appreciate the comments raised by the Reviewer 1 on our manuscript and on the way that our study was designed and written. As the Reviewer 1 mentions, IgAV patients are rare. Accordingly, one of the main strengths of our study is the large size of our cohort, being constituted by 382 Caucasian patients with IgAV. The Reviewer 1 also makes an interesting question regarding the selection of the genes evaluated in our study that we are pleased to address.
The characteristic pathogenic feature of IgAV is the increased synthesis of an aberrant IgA1 [Rheum Dis Clin North Am 2001, 27, 729-749; Clin Exp Rheumatol 2013, 31(1 Suppl 75), S45-51; Curr Opin Pediatr 2008, 20, 163-170]. This leads to increased IgA1 serum levels, circulating immune complexes containing IgA1 as well as increased deposition of IgA1 in different tissues [Curr Opin Pediatr 2008, 20, 163-170]. Interestingly, the increased synthesis of IgA1 is produced by B lymphocytes, suggesting a relevant role of these cells in the pathogenesis of IgAV. In accordance, Rituximab, a B cell-depleting antibody, has been proposed as an effective and safe therapeutic option for adult-onset with IgAV [Arthritis Rheumatol 2018, 70, 109-114], further supporting B cells as crucial players in the pathogenesis of this condition. Moreover, genetic variants involved the B cell biology were previously described as susceptibility factors for other forms of vasculitis [J Rheumatol 2010, 37, 2076-2080; Hum Immunol 2010, 71, 525-529; Nat Genet 2012, 44, 517-521 and Nat Genet 2012, 44, 522-525]. Based on these considerations, we hypothesized that polymorphisms in genes affecting the biology of B cells are crucial in the pathophysiology of IgAV.
Notwithstanding, the genetic background underlying IgAV is complex, and several genetic variants located in different molecular pathways may influence on the susceptibility and severity of the disease [Autoimmun Rev 2018, 17, 301-315]. In line with this and given the pathophysiological similarities between IgAV and IgA nephropathy [N Engl J Med 2002, 347, 738-748; N Engl J Med 2013, 368, 2402-2414], it is plausible to think that polymorphisms located in genes affecting the mucosal immune defense, previously reported as susceptibility loci in IgA nephropathy [Nat Genet 2014, 46, 1187-1196], may also be implicated in the pathogenesis of IgAV. A previous study of our group evaluated the potential role of TNFSF13 (APRIL), TNFSF13B (BAFF) and TNFRSF13C (BAFFR), mucosal immune defense genes involved in later states of B cell differentiation, in the pathogenesis of IgAV [Sci Rep 2021, 11, 11510]. In contrast to IgA nephropathy [Nat Genet 2014, 46, 1187-1196], our data revealed that these genes do not contribute to the genetic network underlying IgAV [Sci Rep 2021, 11, 11510]. However, these results do not exclude the potential implication of other polymorphisms related to mucosal immune defense (such as ITGAX-ITGAM, VAV3 and CARD9) in the pathogenesis of IgAV and further studies assessing this issue would be of potential interest.
Taking into account all these considerations and based on the interesting comments raised by Reviewer 1, we have now included information assessing this issue in the `Discussion´ section of the revised version of our manuscript as follows (page 9):
`….. With respect to this, genes affecting the mucosal immune defense, having an impact on IgA production by plasma cells in mucosa and previously reported as susceptibility loci in IgA nephropathy [51], may also be implicated in the pathogenesis of IgAV and further studies assessing this issue would be of potential interest´.
Consequently, the following new reference has now been added in the revised version of our manuscript as new references 51:
- Kiryluk, K.; Li, Y.; Scolari, F.; Panacherie, S.; Choi, M.; Verbitsky, M.; Fasel, D.; Lata, S.; Prakash, S.; Shapiro, S.; et al. Discovery of new risk loci for IgA nephropathy implicates genes involved in immunity against intestinal pathogens. Nat Genet 2014, 46, 1187-1196.
Consequently, the order of the remaining references has now been modified in accordance in the revised version of our manuscript.
Thanking you for the opportunity of revising the manuscript according to your interesting comments.
With our best wishes,
Reviewer 2 Report
Dear authors,
you investigated several specific polymorphisms of the CD40, BLK and BANK1 genes, known as susceptibility factors for other forms of vasculitis, for their influence on IgAV in a large cohort of Spanish IgAV patients compared to a healthy control group. No effect of the polymorphisms on the risk of developing IgAV was found.
Comment: The findings are sound and seem to point to IgAV as a different entity from other forms of vasculitis. The findings could be interpreted more broadly discussing the potential role of B-cells in this disease. Perhaps, IgAV is not a state of increased B-cell activation. For example, B-cell depletion does not seem to affect IgANephritis.
Therefore, a better explanation of the actual pathophysiolgical significance of the investigated genes in the other forms of vasculitis is suggested in your introduction. Genome-wide studies for IgANephritis, which is pathophysiologically very similiar if not identical to IgAV, have not shown these genes as significant players. In the discussion, a paragraph on the potential significance of your finding for the understanding of IgAV pathophysiology is necessary. Perhaps B-cell activation is not an important player in IgAV.
Author Response
Dear Reviewer 2,
Thank you very much for your feedback.
We have carefully read your interesting comments and modified our manuscript accordingly (please, see below).
In the following point-by-point reply, we have detailed the changes performed, which are shown in red in the revised version of our manuscript.
Point-by-point reply
Dear authors,
you investigated several specific polymorphisms of the CD40, BLK and BANK1 genes, known as susceptibility factors for other forms of vasculitis, for their influence on IgAV in a large cohort of Spanish IgAV patients compared to a healthy control group. No effect of the polymorphisms on the risk of developing IgAV was found.
Comment: The findings are sound and seem to point to IgAV as different entity from other forms of vasculitis. The findings could be interpreted more broadly discussing the potential role of B-cells in this disease. Perhaps, IgAV is not a state of increased B-cell activation. For example, B-cell depletion does not seem to affect IgA Nephritis.
Therefore, a better explanation of the actual pathophysiological significance of the investigated genes in the other forms of vasculitis is suggested in your introduction. Genome-wide studies for IgA Nephritis, which is pathophysiologically very similar if not identical to IgAV, have not shown these genes as significant players. In the discussion, a paragraph on the potential significance of your finding for the understanding of IgAV pathophysiology is necessary. Perhaps B-cell activation is not an important player in IgAV.
Response: We thank the Reviewer 2 for his/her positive comments on our manuscript. Additionally, the Reviewer 2 makes reasonable suggestions that we are pleased to address.
The molecular mechanisms underlying IgAV are complex and intricate [Autoimmun Rev 2018, 17, 301-315]. The CD40, BLK and BANK1 genetic variants evaluated in our study were previously described as susceptibility factors for other forms of vasculitis, supporting the relevance of B cell activation in the pathophysiology of these vasculitides. In particular, CD40 rs1883832 [J Rheumatol 2010, 37, 2076-2080] and BLK rs2736340 [Hum Immunol 2010, 71, 525-529] were identified as susceptibility loci for the development of ischemic manifestations in patients with giant cell arteritis, whereas CD40 rs4813003 [Nat Genet 2012, 44, 517-521], BLK rs2254546 [Nat Genet 2012, 44, 517-521], BLK rs2736340 [Nat Genet 2012, 44, 522-525] and BLK rs2618476 [Nat Genet 2012, 44, 522-525] were identified as susceptibility loci for Kawasaki disease. Interestingly, our findings revealed no effect of CD40, BLK and BANK1 polymorphisms on the risk of developing IgAV, suggesting that IgAV may not be a state of increased B-cell activation, and pointing to IgAV as a different entity from other types of vasculitis. In keeping with our results, and as accurately mentioned by Reviewer 2, genome-wide studies for IgA nephritis, which is pathophysiologically similar to IgAV [N Engl J Med 2002, 347, 738-748; N Engl J Med 2013, 368, 2402-2414], have not identified CD40, BLK and BANK1 genes as significant players in the pathogenesis of the disease [Nat Genet 2011, 43, 321-327; Nat Genet 2011, 44, 178-182; Nat Genet 2014, 46, 1187-1196].
Taking all these considerations into account, and according to the accurate suggestion raised by Reviewer 2, a better explanation of the actual pathophysiological significance of the investigated genes in the other forms of vasculitis has now been included in the `Introduction´ section of the revised version of our manuscript as follows (page 2):
`Interestingly, CD40 [9-13], BLK [7, 14-18] and BANK1 [7, 8, 18, 19] genes are identified as susceptibility loci for several inflammatory diseases. Likewise, CD40 and BLK variants are known as susceptibility factors for different forms of vasculitis, specifically for the development of ischemic manifestations in patients with giant cell arteritis [20, 21] and for Kawasaki disease [13, 22], supporting the relevance of B cell activation in the pathophysiology of both vasculitides.…´.
Accordingly, the order of the references has now been modified in the revised manuscript, corresponding the new references 20, 21 and 22 to the previous references 44, 45 and 46 of the first version of our manuscript.
Consequently, the order of the remaining references has now been modified in accordance.
In addition, and based on the interesting suggestion raised by Reviewer 2, we have now included the following paragraph in the `Discussion´ section of the revised version of our manuscript, discussing the potential significance of our finding for the understanding of IgAV pathophysiology (page 9):
`Our findings suggest that IgAV may not be a state of increased B-cell activation, pointing to IgAV as a different entity from other types of vasculitis. In keeping with our results, genome-wide association studies in IgA nephritis, which is pathophysiologically similar to IgAV [47, 48], have not identified CD40, BLK and BANK1 genes as significant players in the pathogenesis of the disease [49-51]. With respect to this, genes affecting the mucosal immune defense, having an impact on IgA production by plasma cells in mucosa and previously reported as susceptibility loci in IgA nephropathy [51], may also be implicated in the pathogenesis of IgAV and further studies assessing this issue would be of potential interest´.
Consequently, the following five new references have now been added in the revised version of our manuscript as new references 47, 48, 49, 50 and 51:
- Donadio, J.V.; Grande, J.P. IgA nephropathy. N Engl J Med 2002, 347, 738-48.
- Wyatt, R.J.; Julian, B.A. IgA nephropathy. N Engl J Med 2013, 368, 2402-2414.
- Gharavi, A.G.; Kiryluk, K.; Choi, M.; Li, Y.; Hou, P.; Xie, J.; Sanna-Cherchi, S.; Men, C.J.; Julian, B.A.; Wyatt, R.J.; et al. Genome-wide association study identifies susceptibility loci for IgA nephropathy. Nat Genet 2011, 43, 321-327
- Yu, X.-Q.; Li, M.; Zhang, H.; Low, H.-Q.; Wei, X.; Wang, J.-Q..; Sun, L.-D.; Sim, K.-S.; Li, Y.; Foo, J.-N.; et al. A genome-wide association study in Han Chinese identifies multiple susceptibility loci for IgA nephropathy. Nat Genet 2011, 44, 178-182.
- Kiryluk, K.; Li, Y.; Scolari, F.; Panacherie, S.; Choi, M.; Verbitsky, M.; Fasel, D.; Lata, S.; Prakash, S.; Shapiro, S.; et al. Discovery of new risk loci for IgA nephropathy implicates genes involved in immunity against intestinal pathogens. Nat Genet 2014, 46, 1187-1196.
Consequently, the order of the remaining references has now been modified in accordance in the revised version of our manuscript.
Thanking you for the opportunity of revising the manuscript according to your interesting comments.
With our best wishes,
Round 2
Reviewer 1 Report
The authors aequately answered to my points.
Reviewer 2 Report
Dear authors,
thank you for expanding your discussion accordingly.
No more comments.